# Establishment and Its Utility of a Patient-Derived Cell Xenografts (PDCX) Model with Cryopreserved Cancer Cells from Human Tumor

**DOI:** 10.3390/cells14050325

**Published:** 2025-02-21

**Authors:** Ki Yeon Kim, Ji Min Lee, Eun Ji Lee, Daun Jung, Ah-Ra Goh, Min Chul Choi, Sang Geun Jung, Hyun Park, Sohyun Hwang, Haeyoun Kang, Hee Jung An

**Affiliations:** 1Department of Pathology, CHA Bundang Medical Center, CHA University, Seongnam-si 13496, Republic of Korea; superk8787@chamc.co.kr (K.Y.K.); c094002@chamc.co.kr (J.M.L.); a220715@chamc.co.kr (E.J.L.); jhd2800@chamc.co.kr (D.J.); ahragoh@chamc.co.kr (A.-R.G.); blissfulwin@chamc.co.kr (S.H.); hykang@cha.ac.kr (H.K.); 2CHA Advanced Research Institute, CHA Bundang Medical Center, Seongnam-si 13496, Republic of Korea; 3Department of Gynecologic Oncology, CHA Bundang Medical Center, CHA University, Seongnam-si 13496, Republic of Korea; oursk79@cha.ac.kr (M.C.C.); sgoncol@chamc.co.kr (S.G.J.); p06162006@cha.ac.kr (H.P.)

**Keywords:** patient-derived xenografts, primary cancer cell, ovarian cancer, HGSC, cryopreservation

## Abstract

Patient-derived xenograft (PDX) models are powerful tools in cancer research, offering an accurate platform for evaluating cancer treatment efficacy and predicting responsiveness. However, these models necessitate surgical techniques for tumor tissue transplantation and face challenges with non-uniform tumor growth among animals. To address these issues, we attempted to develop a new PDX modeling method using high-grade serous ovarian cancer (HGSC), a fatal disease with a 5-year survival rate of 29%, which requires personalized research due to its morphological, genetic, and molecular heterogeneities. In this study, we developed a new patient-derived cancer cell xenograft (PDCX) model with high engraftment efficiency (64%) that utilizes primary cancer cells instead of patient tissues. Primary cancer cells can be stably cryopreserved for extended periods (up to 485 days), and when transplanted into female NSGA mice, they maintain morphological and molecular characteristics without significant genetic differences compared to their original primary tumors. Furthermore, PDCX models can be easily produced using a syringe, allowing for uniform tumor sizes across multiple animals. Additionally, M2 PDCXs exhibited a significantly faster growth rate compared to M2 PDTXs. Consequently, our PDCX model offers a streamlined approach for evaluating personalized cancer treatments with minimal experimental variability.

## 1. Introduction

Patient-derived xenograft (PDX) models have emerged as valuable tools for cancer research and drug development. These models are established by transplanting human tumor tissue into immunodeficient mice, thereby preserving the cancer microenvironment, heterogeneity, and molecular characteristics of the original tumor [1,2,3]. PDX models offer significant advantages over traditional cell line-derived xenografts, particularly in their ability to more accurately reflect the complexity of human tumors. PDX models play a crucial role in cancer drug screening and development by evaluating drug safety, efficacy, and pharmacokinetic properties in preclinical trials [4,5]. They also allow for the observation of molecular changes that help predict individual treatment responses and improve clinical trial design and patient selection [6]. Therefore, PDX models are powerful tools in cancer research, offering a more accurate and reliable platform for evaluating the efficacy of cancer treatment and predicting its responsiveness in terms of precision medicine.

In this regard, there are several limitations to planning an in vivo experiment using PDX models. First, the engraftment rates of PDX models vary greatly depending on the condition of the primary tumor. Even within the same tumor, if the cellular composition of each piece is different, the growth rate of each xenografted tumor will be different [5]. Even if the tissue fragments have the same condition, it is not easy to cut tumor tissue into numerous small pieces of uniform size. If uneven tissues are used for xenografting, the experimental results will be quite difficult to analyze. Therefore, to ensure equitable grouping, a larger number of mice is required. Another problem here is that xenografting human tumor tissue into mice requires the use of immunodeficient mice, which are more expensive than wild-type mice. Therefore, this increases the cost of producing a PDX model. Second, although it would be best to inject it into mice immediately after extracting the patient’s tumor, most laboratories freeze the tissue because it is difficult to transplant fresh tissue into mice in a timely manner. However, freezing tissue can cause tissue damage due to ice crystal formation, and the viability after thawing is very low [7]; therefore, this could reduce the engraftment rates. Lastly, when producing a PDX model using tumor tissue, surgical techniques such as anesthesia, incision, and suturing are required to transplant it into an animal.

Meanwhile, high-grade serous ovarian carcinoma (HGSC) is the most aggressive and prevalent subtype of epithelial ovarian cancer, accounting for approximately 70% of cases, with 85% diagnosed at stage 2 or higher, leading to a 5-year survival rate of only 29% for late-stage diseases [8,9]. Despite being initially platinum sensitive, 20–30% of patients do not respond to chemotherapy, with recurrence frequently occurring within six months post-treatment, underscoring the limitations of current standard therapies [10]. Characterized by significant genetic and molecular heterogeneities, HGSC has exhibited a wide range of copy number alterations (CNAs) and multiple molecular subtypes [11,12], which complicated treatment development, necessitating personalized approaches. Therefore, in this study, we attempted to develop a more advanced PDX model for this type of tumor.

In general, for establishing PDX models, the method of freezing tumor samples has been used for a long time, but its success rate has been low. The first PDX model using cryopreserved primary cancer tissue showed a transplant success rate of only 21% [13]. To overcome this, a study published in 2015 increased the transplant success rate to 68% by minimizing ice crystals using the vitrification method, but it was cumbersome because it required three steps for tissue freezing and four steps for thawing [14]. In other studies, instead of using tissue fragments, cancer cells isolated from a patient’s tumor have been used to simply generate PDX models [15,16,17]. Some experiments reported that sufficient numbers of patient-derived cell xenograft (PDCX) mice were established with as few as 2 × 10^5^ cells after one or two passages in vivo [16]. However, in most of the reports covering PDCXs, the cancer cells were injected into mice immediately after isolation from a tumor. Therefore, the experiments must have been laborious, as many steps had to be completed immediately within a short period of time.

In this study, we established a method for the PDCX model with cryopreserved cells from HGSCs. In this process, we isolated cells from tumor tissues of HGSC patients and cryopreserved them in LN_2_, and then the cells were used to establish PDCX models using a non-surgical method as needed. We found that the tumors of PDCX mice generated in this manner were histologically and genetically identical to those of original human tumors or patient-derived tissue xenograft tumors (PDTXs). Additionally, we showed that uniform-sized tumors were formed in the third generation PDCX model using a small number of cells.

## 2. Materials and Methods

### 2.1. Tumor Tissue Samples from Patients

All the HGSC tissue specimens were obtained from 14 ovarian cancer patients who underwent tumor removal surgery via salpingo-oophorectomy at CHA Bundang Medical Center from October 2019 to March 2023 (Table 1). None of the patients received pre-operative chemotherapy. The tissues were transported to the laboratory immediately after surgery in cold saline solution and stored at 4 °C until further processing but no longer than 1 day. The study was approved by the institutional review board of CHA Bundang Medical Center (IRB #CHAMC 2019-08-039) and was conducted with written informed consent from all patients.

### 2.2. Mouse and Study Design

Six-week-old NSGA (NOD.Cg-Prkdcscid Il2rgtm1Wjl/SzJ) female mice (JA BIO, Inc., Suwon, Republic of Korea) were used after one week of adjustment. The mice were maintained in a specific-pathogen-free (SPF) animal facility at CHA Advanced Research Institute, CHA Bundang Medical Center (Seongnam, Republic of Korea) under a temperature of 24 ± 3 °C and a 12-h light and dark cycle, with unrestricted access to food and water. After the acclimatization period, mice from all cages were randomly mixed and used. The sample size was determined based on practical constraints and the availability of patient-derived cancer cells or tissues. No formal a priori sample size calculation was performed due to the exploratory nature of the study and the variability in the available patient-derived or M1 tumors. When sufficient cancer cells were available for M2 or M3 models, 2–7 mice were used per group. We excluded data analysis in cases where tumor formation was insufficient.

For M1 PDCX modeling, we transplanted primary cancer cells to 1–4 mice per case. A total of 36 mice were used for 14 cases, including 2 cases of primary tissue-transplanted PDTX models (SC 155 and SC 156) for comparison with the PDCX model. For M2 PDCX and PDTX production (Appendix A), 14 mice were used (SC155 PDCX, 2 mice; SC156 PDCX, 7 mice; SC156 PDTX, 3 mice; SC214 PDCX, 2 mice). For M3 PDCX production, 9 mice were used (SC156 PDCX, 5 mice; SC214 PDCX, 4 mice).

Each step of the PDCX production process is detailed in the subsections below, but can be briefly summarized as follows:Obtain a tumor from an HGSC patient.Isolate cancer cells from the patient’s tumor tissue using collagenase A and DNase I.Cryopreserve the isolated cancer cells using Gibco freezing agent (Gibco, Cat# 12648010, Grand Island, NY, USA). When PDCX production is required, rapidly thaw the cells using a 37 °C water bath.Wash the thawed cells twice with cancer media, count, and adjust to the desired cell number/75 μL cancer media. Mix with an equal volume of Matrigel, load into a 1 mL syringe, and maintain at 4 °C.Restrain the immunodeficient mouse for PDCX production, remove hair from the injection site, and disinfect with an alcohol swab.Inject 150 μL of cell-containing media with Matrigel subcutaneously.

### 2.3. Single-Cell Isolation of Primary Cancer Tissue

The fresh tumor tissue obtained from HGSC patient was placed in a 100 mm culture dish (SPL, Cat# 20100, Pocheon, Republic of Korea), containing McCoy’s 5A medium (Gibco, Cat# 16600-082, Grand Island, NY, USA). Fat and contaminants contained in the tissue were removed using fine forceps and washed two to three times with the media. After transferring 3 g of tumor tissue to a 50 mL tube (SPL, Cat# 50050, Pocheon, Republic of Korea), 10 mL of McCoy’s 5A medium was added, and the tissue was finely cut into small pieces. Some small pieces of tissue were snap-frozen immediately to create the PDTX model, while others were dissociated into cells for the PDCX model. When freezing the specimen in tissue form, one of the tissue fragments was simply minced to approximately 1 mm^3^. To isolate the specimen into cancer cells, we used the Miltenyi gentleMACS Tissue Dissociator (Miltenyi Biotec, Cat# 130-093-235, Bergisch Gladbach, Germany) with the Multi Tissue Dissociation Kit (Miltenyi Biotec, Cat# 130-110-201, Bergisch Gladbach, Germany). Tissue samples were placed in a c-Tube with the enzymatic dissociation reagents and processed according to the gentleMACS program, with 30-min incubation at 37 °C. The cell suspension was filtered through a 70 μm MACS SmartStrainer (Miltenyi Biotec, Cat# 130-098-462, Bergisch Gladbach, Germany), centrifuged at 1500 rpm for 10 min, and then resuspended in McCoy’s 5A medium containing 10% Fetal Bovine Serum (FBS) (Gibco, Cat# 26140-079, Grand Island, NY, USA) for cell counting.

### 2.4. Cryopreservation and Thawing

After counting, the cells were centrifuged at 1500 rpm for 5 min, the supernatant was discarded, the cells were resuspended in Gibco freezing agent, and 1 × 10^7^ cells were frozen in 1.8 mL cryovials (Thermo Scientific, Cat# 377267, Roskilde, Denmark). All the cryotubes were transferred to a −1 °C/min freezing container, slowly lowered to −80 °C, and then transferred to liquid nitrogen for long-term storage. For thawing, the tubes were quickly thawed in a 37 °C water bath for 2 min. Each ml of thawed cells was transferred to a clean tube containing 10 mL of cancer medium with 20% FBS. After washing via centrifugation (1600 rpm, 10 min, RT), cancer medium was added to the pellet, and the cell numbers were determined.

### 2.5. Cancer Cell Isolation from M1 or M2 Tumors of PDCX or PDTX Mice

M1 or M2 tumors of PDCX or PDTX mice were extracted when they grew to over 500 mm^3^ in the volumes. The tumor tissues were placed in 60 mm Petri dishes (SPL, Cat# 10060, Pocheon, Republic of Korea) with dissociation buffer composed of Hanks’ Balanced Salt Solution (HBSS, Gibco, Cat# 14175-095, Grand Island, NY, USA) with 2 mg/mL Collagenase A (Roche, Cat# 10103578001, Mannheim, Germany) and 27 µg/mL DNase I (Sigma-Aldrich, Cat# D5025, St. Louis, MO, USA). The tumor tissue was cut into small pieces using a fine razor and kept at 37 °C for the enzymatic reactions for 20 min. When the tissue was sufficiently decomposed, EDTA (Gibco, Cat# 15400-054, Grand Island, NY, USA) was added to 0.5 mM to stop the enzyme reaction and filtered through a 70 µm cell strainer (SPL, Cat# 93070, Pocheon, Republic of Korea). The tube was filled with complete cancer medium (MacCoy’s 5A with 10% FBS and 1% antibiotic–antimycotic (Gibco, Cat# 15240-062, Grand Island, NY, USA), washed via centrifugation (1800 rpm, 10 min, RT), and the remaining pellet was resuspended in 1X RBC lysis buffer (Biolegend, Cat# 420302, San Diego, CA, USA) to remove red blood cells. After 15 min, samples were washed twice with complete medium (1500 rpm, 5 min, RT) and resuspended in the medium for cell counting.

### 2.6. In Vivo Transplantation

The cryopreserved cells were thawed, counted, and adjusted to a maximum of 1 × 10^7^ cells/75 µL/mouse. For the cryopreserved tissue fragments, the tissue pellet was adjusted to 150 mg or 160 mg/mouse. Both tissue and cell processing was performed under the same conditions and on the same dates, and there was no consistency in the order of tumor measurement. An equal volume of Matrigel (Corning, Cat# 454248, Corning, NY, USA) was mixed at 4 °C and injected into the flank of the mice using a 1 mL syringe. For tissue fragment transplantation, 150 mg or 160 mg of tissue fragments were mixed with Matrigel and placed in a 1 mL syringe with an 18G needle. After anesthetizing the mouse, an incision of approximately 0.5 mm was made on the flank, the tumor tissue was injected, and the incision site was sutured with a clip. No unexpected adverse events occurred during the mouse experiments. Tumors were measured once a week using calipers, with the volume calculated as follows: Tumor volume = Width^2^ × Length × 0.5.

The endpoint of the mouse experiments was when the tumor volume reached approximately 1000 mm^3^ or when the xenografted mice did not develop tumors within 6 months.

### 2.7. Specific Growth Rate (SGR) Method

The Specific Growth Rate (SGR) was calculated to standardize tumor growth across different initial tumor volumes and experimental durations. The SGR value per 100 days was determined using the following equation:SGR = [ln(V2) − ln(V1)]/(t2 − t1) × 100
where V1 and V2 are the tumor volumes or implanted tumor cell numbers at the initial time point (t1) and the final time point (t2), respectively.

### 2.8. Immunohistochemistry

The extracted tumors from PDX mice were cut into small pieces and fixed in 4% formalin for at least 1 day. The tissue was prepared for a paraffin-embedded block. Four μm sections of formalin-fixed, paraffin-embedded blocks were placed on amino-propyl-ethoxy-silan-coated glass slides. The sections were stained with hematoxylin and eosin (H&E) and stained with anti-human antibodies to detect p53, PAX8, Ki67, vimentin, and anti-mouse CD31 (Cell signaling Technology, Cat# 77699T, Danvers, MA, USA). Except anti-mouse CD31, immunohistochemical stains were performed with a Ventana Benchmark automated staining platform (Ventana Medical Systems, Incl, Tucson, AZ, USA) using a VENTANA OptiView diaminobenzidine (DAB) tetrahydrochloride IHC Detection Kit (P/N 760-700) and its staining protocol. The sections were deparaffinized, rehydrated, and heated following the protocol. The antigen retrieval time, retrieval buffer, and incubation time of each antibody are shown in Table 2. For the anti-mouse CD31 staining, the procedure was performed by CM BioPATH, a preclinical pathology contract research organization based in Republic of Korea. Briefly, using the ABC kit (Vector, Cat# PK-6100, Burlingame, CA, USA), the incubation condition of primary antibody is detailed in Table 2. Secondary staining was conducted using anti-rabbit HRP (Dako, Cat# K4003, Agilent Technologies, Santa Clara, CA, USA) for 30 min. The investigators of this experiment were unaware of the details of the study.

### 2.9. DNA Isolation

For genetic analysis, DNA was extracted from the cryopreserved cells that were isolated from SC155 and SC156 patients’ tumors and from their corresponding M1 and M2 PDCX tumors. The extraction was performed according to the standard protocol. After thawing the cancer cells in a 37 °C water bath, the cells were washed with DPBS via centrifugation (1600 rpm, 5 min, RT), and the supernatant was discarded. DNA was extracted from the pellets using DNeasy Blood & Tissue Kits (Qiagen, Cat#69504, Hilden, Germany). The quality and quantity of the DNA samples were assessed using a NanoDrop (Lifereal, Cat# FC-1100, Hangzhou, China) and delivered to a genetic analysis company.

### 2.10. Genetic Analysis

The genetic analysis was performed by DNA Link Inc (Seoul, Republic of Korea). An Agilent SureselectXT V6 kit was used for Whole-Exome Sequencing (WES) analysis. Briefly, mouse contamination reads were removed from the produced data using the bbsplit tool of bbmap. Kinship analysis was conducted after converting the variant calling format (VCF) to the plink genome (PLINK). The heatmap was drawn using the PI_HAT value obtained at this time. Allele-specific copy number analysis of tumors (ASCAT) was performed using the copy number alterations (CNAs). The CNAs on all the chromosome loci are shown.

For Kinship analysis, VCF files from GATK’s HaplotypeCaller for 6 samples were merged using the PLINK program. Genotype matching (Z0, Z1, and Z2) was calculated through pairwise analysis using the following command: plink --bfile merged.vcf --genome --min 0.3 --out merged --allow-no-sex --threads 80. PI_HAT values, distributed between 0 and 1 (with values closer to 1 indicating a higher likelihood of genetic identity), were displayed as a heatmap.

Allele-specific copy number analysis of tumors (ASCAT) was employed to analyze copy number alterations in tumor samples. This algorithm utilizes matched tumor–normal sample pairs to accurately determine allele-specific copy number profiles while accounting for tumor purity and ploidy. The detailed processes are as follows: Sample preparation and data extraction: Tumor and matched normal samples were processed to obtain Log R Ratio (LogR) and B Allele Frequency (BAF) data. These metrics serve as inputs for the ASCAT algorithm, providing information on the total copy number and allelic imbalance, respectively. GC content correction: A GC-wave correction was applied using the matched normal samples to mitigate biases associated with GC content and replication timing. This step enhances the accuracy of subsequent copy number estimations. Segmentation: The LogR and BAF data from both tumor and normal samples underwent simultaneous segmentation. This process identifies regions of constant copy numbers across the genome, taking into account both samples to improve breakpoint detection. Tumor purity and ploidy estimation: ASCAT estimated the tumor purity (fraction of tumor cells) and ploidy (average number of copies per cell) using the segmented data. The matched normal sample served as a reference, allowing for more precise estimations of these critical parameters. Allele-specific copy number calculation: By comparing the tumor sample to the matched normal, ASCAT computed allele-specific copy numbers for each segment. This approach enables the identification of tumor-specific alterations while filtering out germline variants.

The investigators of this experiment were unaware of the details of the study.

### 2.11. Statistical Analysis

Statistical analyses were performed using GraphPad Prism software (GraphPad Software, Inc., San Diego, CA, USA). We used a *t*-test, simple linear regression, or one-way ANOVA test for the comparison between groups. *p*-values < 0.05 were considered significant. To compare PDCXs and PDTXs using their SGRs, the Bland–Altman analysis function of GraphPad Prism V.8.0.2 was additionally used.

## 3. Results

### 3.1. Cryopreserved Primary Cancer Cells Can Be Used to Establish a PDX Model for HGSC

To determine whether cryopreserved cells isolated from tumors of human high-grade serous carcinoma (HGSC) would establish a PDX model when implanted in immune-deficient mice, fresh tumor tissues from HGSC patients were obtained from CHA Bundang Medical Center. To generate the PDCX models, 2 to 10 × 10^6^ of the cryopreserved cancer cells for each case were inoculated in the flank of immune-deficient NOD-Prkdc^em1Baek^Il2rg^em1Baek^ (NSGA) mice (Figure 1A). Nine out of fourteen cases (64%) were successfully established tumors on mice (mouse passage 1; M1) (Table 3). When the established M1 tumors were compared with the corresponding primary tumors using H&E staining, all the cases (nine cases) showed consistency with the primary tumors (Figure 1B). These results showed that cryopreserved cells from patients’ tumor tissues are a good material for PDX modeling.

To check whether the engraftment rate decreased as the cryopreservation period increased, the cases were divided into three period groups: short (under 100 days), middle (between 100 days and 1 year), and long (over 1 year). The engraftment rates were 73.3% in the short term, 50% in the middle term, and 50% in the long term (Appendix A). Interestingly, the long-term group (SC39 and SC214) showed high engraftment rates, like that of middle period group, although their engraftment rates slightly decreased compared to the cases from the short period. This result proved the stability of cryopreserved cancer cells for PDCX establishment after long-term storage, such as over 1 year. In addition, when SC214 was tested for PDCX engraftment with both short-term (24 day) and long-term (386 days) groups, the engraftments were successful with both periods. These findings suggest that the cryopreserved primary cancer cells maintain their viability for extended periods, potentially up to and beyond one year. This observation has significant implications for long-term biobanking and offers flexibility in experimental design and execution.

### 3.2. Comparison of Tumor Growth Rate and Tumor Size Between PDCX and PDTX Models

To establish PDCX modeling, it is also required to check whether there is any difference between patient-derived cell xenografts (PDCXs) and patient-derived tissue xenografts (PDTXs). In two cases (SC155 and SC156), we dissociated part of the tumor into cells using enzymes and simply chopped the other part into small tissues and then cryopreserved it. The dissociated cancer cells for PDCXs and the tissue fragments for PDTXs were thawed just before inoculation. In each case, 5 × 10^6^ cancer-derived cells or 150 mg of tumor tissues were inoculated on the right flank of three to four NSGA mice. The PDCX group showed a fairly uniform tumor growth rate, while the PDTX group showed notable individual variations (Figure 2A, right) or no growth at all (Figure 2A, left). In the case of SC156, the transplantation success rate was 100% for both the PDCX and PDTX, whereas in the case of SC155, the success rate was 50% for the PDCX and 25% for the PDTX. Initially, the SC155 PDTX model showed tumor formation in two mice; however, these tumors did not exhibit growth grossly until day 140. After 193 days, when the remaining mice were sacrificed, a very small tumor tissue weighing only 180 mg was unexpectedly obtained from one mouse. Unfortunately, it was too small to be used for the next passage. In terms of the time it took for the tumors to reach 500 mm^3^, it took 56.5 days for the SC156 PDCX and 64.3 days for the SC156 PDTX (Figure 2B).

Considering that 3.2 × 10^8^ cancer cells were obtained from 11.2 g of SC155 primary cancer, and 1.7 × 10^8^ cancer cells were obtained from 6.4 g of SC156 primary cancer (Table 1), 1 g of tumor tissue would be dissociated into 2.7 × 10^7^ cells. This means that 150 mg of primary cancer tissue corresponds to approximately 4 × 10^6^ cells. Therefore, the PDTX injected a slightly smaller amount (80%) of primary tumor cells compared to the PDCX, which injected 5 × 10^6^ cells. However, the PDTX showed a significantly lower engraftment rate and tumor growth rate compared to the PDCX, suggesting that primary tumors in a cell state are more efficient than in a tissue state.

When the M1 tumor volumes were sufficiently large, greater than 500 mm^3^, the tumors were extracted, and tumor cells and tissues were prepared and frozen in the same manner as before. For generating M2 (mouse passage 2) mice, seven SC156 PDCX_M2 mice were produced from three SC156 PDCX_M1 tumors, and two SC156 PDTX_M2 mice were produced from one SC156 PDTX_M1 tumor (Table 4, Figure 2C). At this time, the SC156 PDCX_M2 mice were produced using 7 to 8 × 10^5^ M1 tumor cells, with some experiments utilizing half that number (3.5 to 4 × 10^5^ cells) to assess tumor growth at lower cell densities. Concurrently, SC156 PDTX_M2 was established using 160 mg of M1 tumor tissue. To maximize the number of M2 tumor cells using a small number of animals, we inoculated both the left and right flanks of the mice this time. The M2 PDTX model produced tumors ranging from 720 to 880 mg at 87 days after implanting 160 mg of tumor tissue. In contrast, the M2 PDCX model, initiated with only 7 × 10^5^ cells, yielded tumors of similar or greater mass (670–1140 mg) at 80 days (Table 4). Due to variations in the initial tumor volume and extraction dates among individual mice, we employed the Specific Growth Rate (SGR) method to compare tumor growth rates between PDCX and PDTX models. This method calculates the daily percentage increase in tumor volume, assuming exponential tumor growth, thereby allowing for standardized comparisons across different experimental time frames and initial tumor sizes [18]. When calculating the SGR over a 100-day period for each tumor, we observed that M2 PDCXs exhibited a significantly faster growth rate compared to M2 PDTXs (5.04 vs. 1.85, **** *p* < 0.0001) (Table 4 and Figure 2D). To further contextualize our findings, we conducted a Bland–Altman analysis using GraphPad Prism to compare tumor growth rates between PDCX and PDTX models. Due to variations in implantation quantities, we compared Specific Growth Rate (SGR) values (Appendix A). The Bland–Altman method calculates the mean difference between two measurement methods (bias) and 95% limits of agreement as the mean difference ± 1.96 standard deviations [19]. We plotted the difference between paired PDCX and PDTX measurements (Y-axis) against their average (X-axis). Our analysis revealed that the 95% limits of agreement did not include 0 (range: 1.591–5.129), indicating a systematic difference between the methods. The graph showed a positive correlation between the average SGR and difference, suggesting inconsistency between the PDCX and PDTX models across the measurement range. This analysis complements the SGR comparison in Figure 2D, providing additional context for understanding our cell implantation method’s efficiency compared to fresh tissue implantation. Therefore, it is thought that the PDCX has a greater efficacy than the PDTX in the M2 PDX compared to in the M1 PDX. In addition, these results also demonstrate that even a small amount, such as 3.5 × 10^5^ cells, of cryopreserved M1 cancer cells is sufficient to generate M2 PDCX models.

When we compare the growth rate between M1 and M2 tumors, the M2 tumors grew approximately twice as fast as M1 tumors regardless of whether they were PDCXs or PDTXs (Figure 2E). To calculate the number of cells obtained from an M2 tumor, the tumors were dissociated into cells. The live cells obtained from SC155 and SC156 averaged 5.57 × 10^7^ cells per 1 g (5.5 × 10^7^ from SC155 and 5.6 × 10^7^ from SC156) (Appendix A). Taken together, the PDCX models exhibit higher engraftment rates and faster tumor growth rates compared to the PDTX models, and the M2 tumor growth rate was twice that of the M1 tumor.

### 3.3. M1 and M2 Generations of PDCXs and PDTXs Are Histologically and Immunohistochemically Conserved in Their Originalities

To confirm whether tumor tissue xenografted over several generations maintains its original histological properties, the morphology and protein markers of the available samples from each generation were analyzed in PDCX and PDTX tumors of SC155 (n = 2 and n = 1, respectively) and SC156 (n = 4 and n = 3, respectively). Despite the differences in tumor size depending on the amount of injected cancer cells or tissue and the time of extraction, the M1 and M2 tumors extracted from PDCX and PDTX models showed similar appearances when visually inspected (Figure 3A). By examining the H&E-stained cross-sections of PDCX and PDTX tumors, we determined that there were no morphological differences across all generations and no significant differences compared to the primary tumor (Figure 3B,C).

Regarding the immunohistochemical stains for protein markers, the most important aspect of producing a PDX model is ensuring that the expression of p53 in the PDX model mirrors that of the original tumor. The abnormal expression of p53, indicating a mutation in the p53 tumor suppressor gene, is a common characteristic in various types of cancer, including HGSCs. This abnormal expression can indicate the severity of tumor progression and prognosis [20]. When comparing p53 expression in the PDCX and PDTX models with the original tumors, the immunohistochemical staining results showed consistent patterns with the original tumors in both cases, even after several generations (Table 5, Figure 3B,C). Specifically, all SC155 tumors exhibited a null (−) pattern, while all SC156 tumors showed a 100% strong positivity pattern.

When we checked the expression of PAX8 in the two generations of PDCX and PDTX tumor tissues, it was well preserved, the same as in the original primary tumors. Additionally, the high levels (70~90% positivity) of Ki67 in the SC155 case on the xenografted tissues showed the maintenance of a high proliferation rate (80~90%) over multiple generations. On the other hand, the Ki67 index in SC156 on the xenografted tumor showed a higher proliferation index (80~90%) compared with the moderate Ki67 index (15~25% positivity) of their primary tumor. In M1 and M2 xenograft tumors, the expression of the human stromal marker, vimentin, was focally present in the stroma and blood vessels in some PDCX tumors. The mouse endothelial CD31 was positively stained in the blood vessels of M1 and M2 tumors in the PDCX and PDTX models, indicating that the tumors were adapting to the mouse environment over several generations.

These results showed that xenografted tumors produced using frozen cells are not different from xenografted tumors using tissue and that they adapt to the mouse environment as generations pass but maintain the morphological characteristics of the primary tumor, except the increased Ki-67 index in one case.

### 3.4. PDCX and PDTX Tumors Maintain Genetic Conservation from the Original Tumors Across Multiple Generations

In PDX model production, genetic testing between the xenografted tumor and the primary tumor is essential to evaluate the accuracy of the model. To confirm if the tumors from M1 and M2 PDCXs preserve the genetic characteristics descended from their corresponding patients’ primary tumors, a WES microarray was performed using M1 and M2 PDCX tumors and their primary tumors for two cases (SC155 and SC156).

Kinship analysis showed that the PI_HAT (Proportion IBD) value between each patient’s primary tumor cells and their corresponding M1- and M2-derived tumor cells exceeded 0.91, confirming the genetic identicality and ruling out potential sample cross-contamination or misidentification (Table 6). When these results were analyzed using a heatmap, the original patients’ tumor cells and the two generations of mouse tumors derived from them were clustered together (Figure 4A). In contrast, the PI_HAT values between the SC155 and SC156 samples, derived from different patients, were significantly lower, approximating 0.7, indicating genetic distinctness and confirming their origins from different individuals. These results show that even if multiple generations of PDXs are produced using cryopreserved cells, the tumor tissue retains the same human characteristics.

Additionally, ASCAT analysis was used to identify the size and location of CNAs in M1 and M2 tumor cells (Figure 4B,C). In SC155M1 and M2, chromosomes 1, 6, 7, 10, 11, 12, 16, and 19 exhibited recurrent CNAs. Similarly, SC156M1 and M2 displayed frequent CNAs on chromosomes 1, 3, 6, 11, 15, 16, 19, and 22. The preservation of these CNA patterns between M1 and M2 passages for each sample indicates the retention of intrinsic genetic characteristics.

### 3.5. Third Generation of PDCXs Can Efficiently Be Used in Large-Scale Tests with Uniform Tumor Growth

In experiments using animals, inter-individual variation is bound to occur. To reduce this variation and ensure accurate evaluation, it is recommended that experiments use at least five mice per group. In animal experiments using PDTXs, it is difficult to inject the same amount of tissue pieces into many subjects, so the results have to be derived from a larger number of subjects or considering that tumors grow differently. However, if the PDCX model is used, it is possible to reduce inter-individual variation by injecting the same number of cells. To prove this, we administered 2 × 10^6^ M2 cancer cells obtained from the SC156 M2 PDCX and SC214 M2 PDCX into NSGA mice to produce M3 PDCX mice and observed tumor growth. To prove that the tumor growth rates were uniform among the entities, the ordinary one-way ANOVA analysis function of the GraphPad Prism program was implemented. This method is for determining the statistical significance of differences between groups. Specifically, differences are considered significant when the *p*-value is less than 0.05 and the R-squared value approaches 1. Conversely, minimal differences between subjects are indicated by *p*-values approaching 1 and R-squared values close to 0. This interpretative framework allows for a nuanced assessment of inter-group variability. When the tumor growth curve of each mouse was analyzed with this method, the *p*-value was 0.6461 for SC156M3 and 0.9864 for SC214M3. In addition, the R-squared values were 0.02958 and 0.002698, respectively (Figure 5). These results confirmed that PDCX model markedly reduce the inter-individual variation by administering the same number of M2 cancer cells to each individual, and it is possible to produce multiple M3 PDCXs with the same conditions, which is more suitable for in vivo anticancer efficacy evaluation.

## 4. Discussion

Patient-derived xenograft (PDX) models have significantly advanced cancer research by providing more accurate representations of human tumors in preclinical settings. These models involve transplanting patient tumor tissue directly into immunodeficient mice, preserving tumor heterogeneity and the microenvironment. PDX models have shown varying engraftment rates across cancer types, with colorectal cancer at 76%, head and neck cancer at 85%, non-small-cell lung cancer at 41%, breast cancer at 19–21%, gastric cancer at 24%, and liver cancer at 14%. While valuable for drug discovery and personalized medicine, traditional PDX models face challenges such as low engraftment rates for certain cancers and scaling difficulties. High-grade serous carcinoma (HGSC) exhibits a relatively high engraftment rate of 58% [21]. However, the inherent high heterogeneity of HGSC increases the likelihood of different tumor phenotypes developing in different mice when utilizing tissue slices, thereby diminishing inter-subject uniformity in experimental design [4,22]. In addition, beyond HGSC, the method of creating PDXs using primary tumor tissue requires surgical techniques such as anesthesia, incision, and suturing, so it may be difficult to start in laboratories where these techniques are not established. Additionally, these surgical techniques cause significant stress to mice.

In the present study, we demonstrated that the PDCX model has an advantage for advancing in vivo cancer research, particularly in the context of high-grade serous carcinoma (HGSC). This innovative approach offers several key advantages over traditional patient-derived xenograft (PDX) models. The first one is that it shows successful tumor engraftment potential with long-term viability. The traditional PDTX engraftment rates vary significantly depending on cancer type, ranging from 14% for liver cancer to 85% for head and neck cancer when using NSG or NOG mice [23]. Although HGSC has a relatively high engraftment rate, with reported success rates of 58% at 3–6 months [21,24], the authors have suggested the complex preparation method of using fresh tissue. In our study, the PDCX model achieved a high success rate of 64% (9 out of 14 cases) using cryopreserved tumor cells with a maximum latency time of 6 months. This result even surpasses PDTX models from previous reports [21,24] using fresh tissue. Although the number of participants was limited, our study demonstrated successful engraftment rates in different cryopreservation periods. As shown in Appendix A, the engraftment rate was 73.3% in the short period (under 100 days), 50% in the middle period (between 100 days and 1 year), and 50% in the long period (over 1 year). Notably, two cases, SC39 and SC214, exhibited tumor growth consistent with the original patient tumors after 485 days and 386 days, respectively. This indicates that cancer cells can be stably preserved for over a year in liquid nitrogen for generating PDCX models. Compared to a published study that reported an only 9% engraftment rate with cryopreserved tissues stored over 52 weeks [25], the PDCX model from our study revealed a significantly high engraftment rate in long-term stored specimens. The second one is that our PDCX modeling simplifies the establishment process, requiring only a syringe without complex surgical techniques that could be burdensome to the mice. In addition, the use of isolated cancer cells can more easily and accurately measure the amount of transplanting material, which results in more uniform engraftment across animals, reducing experimental variability and minimizing outliers. This uniformity is particularly beneficial, especially when using expensive immunodeficient mice, providing economic advantages in research settings.

This study also provided detailed information about the number of cells obtained from extracted M2 tumors. On average, approximately 5.57 × 10^7^ live cells were isolated per gram of M2 tumor tissue. This high yield of cells from the tumor tissue enabled us to develop enough subsequent generations of xenografts. Theoretically, assuming an inoculation dose of 3.5 × 10^5^ cells per animal, 1 g of M2 tumor tissue is sufficient to establish approximately 160 M3 PDCX mice (Table 4). In this study, the cell viability after extraction was consistently high, ranging from 79.7% to 90.9%, indicating the robustness of the isolation process. These findings highlight the scalability of the PDCX approach, enabling researchers to produce large numbers of uniform xenografts from a single tumor sample, which is particularly valuable for large-scale drug screening studies.

This study demonstrated that the PDCX tumors we developed maintained the genetic and histological characteristics of the original patients’ primary tumors. Kinship analysis showed high PI_HAT values (over 0.91) between patient tumor cells and derived M1 and M2 tumor cells, confirming their genetic similarity. ASCAT analysis revealed consistent CNA patterns across generations. Histologically, the expression of key markers, such as p53 and PAX8, was preserved in M1 and M2 PDCX tumors as well as their similarity in H&E morphology to their primary tumors, indicating the maintenance of important tumor characteristics. It was reported that nearly all HGSC cases (96%) display mutations in the TP53 gene, which is crucial for cell cycle regulation, apoptosis, and DNA repair [26,27], while PAX8, a gene that promotes cancer cell proliferation and invasion, is often overexpressed in ovarian cancers, including HGSCs, which serves as an important diagnostic indicator for ovarian cancers [28,29,30,31]. In our PDCX model, we confirmed that M1 and M2 tumors preserved the aberrant expression of p53 protein (null negativity for SC155 and diffuse strong positivity for SC156). In this study, we found that one (SC155) of the PDCX tumors showed a similar Ki67 proliferation index to that of its primary tumor, while another PDCX tumor (SC156) revealed an increased Ki67 proliferation index compared to its primary tumor (80% vs. 25%). This indicates that the PDX model may not always be identical and can be more aggressive. This phenomenon can be attributed to several factors: (1) selection bias: the process of establishing a PDX model might inadvertently select more aggressive tumor cells because the primary tumor may have different aggressiveness in different areas; (2) tumor microenvironment and immune system interaction of the host animal; the tumor micro- and immune environment in the host animal can differ significantly from that of the human patient, potentially promoting faster tumor growth and higher proliferation rates. However, other molecular and genetic characteristics were the same as the primary tumor in SC156, leading us to conclude that the PDX tumors accurately represent the primary tumor.

As for the stromal marker, vimentin, some of PDCX or PDTX tumors showed focal positivity in the stroma and blood vessels. In general, changes in the expression of human stromal markers and mouse endothelial markers were observed across multiple generations of xenografts. The expression of the human stromal marker, vimentin, decreased in M1 and M2 xenograft tumors, while the mouse endothelial marker CD31 increased. This shift indicates that the tumors were adapting to the mouse environment over successive generations [14,32]. The increase in mouse CD31 expression suggests a process of tumor vascularization using the host’s endothelial cells, which is crucial for tumor growth and survival. Taken together, despite some changes in the tumor microenvironment, our xenografts preserved the key morphological and genetic characteristics of the primary tumors, highlighting the robustness of the PDCX model for cancer research. Therefore, the PDCX model in this study holds promise for various applications beyond HGSC in preclinical trials, including cancer drug screening, biomarker discovery, and the evaluation of drug efficacy, safety, and pharmacokinetics. Future studies are needed to focus on expanding the application of this model to other cancer types and exploring its potential for personalized medicine approaches.

The PDCX model offers several significant advantages over traditional PDTX models, particularly in the context of personalized cancer research. Cancer cells can be preserved for extended periods, allowing researchers to initiate animal experiments at their convenience. Notably, a small number of cells can generate numerous PDCX animal models, streamlining the experimental process. While conventional tissue-based methods often burden both researchers and animals with complex modeling procedures, PDCX enables the rapid establishment of multiple animal models using a simplified approach. These benefits position PDCX as a valuable tool for future personalized tumor model research. Compared to PDTX models, PDCX provides opportunities for more extensive experimentation and faster result acquisition. The uniformity in tumor size across PDCX models facilitates the simultaneous analysis of various anticancer drugs, potentially expediting the prediction of patient-specific effective treatments.

Our PDCX models could be widely employed for various research applications previously conducted with PDTX models, including drug screening, preclinical studies for therapeutic effects, and drug resistance mechanisms. Based on the available literature, we can address this concern as follows: (1) In vivo prediction potential: PDCX models, like PDX models, can reproduce drug responses observed in patients. For example, a study showed concordant effects of 5-fluorouracil and radiotherapy in colorectal cancer patients and PDX models [4]. (2) In vitro prediction potential: While our current study focuses on in vivo applications, PDCX models could be adapted for in vitro drug sensitivity predictions, similar to recent successes with patient-derived organoids and cell cultures for drug screening [33]. However, when conducting efficacy evaluations using the PDCX model, it is crucial to first confirm whether the effect of the anticancer drug on the relevant cancer type is also observed in the PDCX model, ensuring the model’s reliability and relevance for specific cancer types and treatments.

In conclusion, we developed a PDCX model that represents a significant advancement in cancer research methodology. In particular, the model’s ability to produce uniform tumor growth in third-generation xenografts (M3 PDCXs) suggests its potential for large-scale, standardized drug efficacy testing with reduced inter-individual variation. In addition, its advantages in terms of long-term storage, uniform tumor growth, easy-to-establish method, and preservation of original tumor characteristics provide a valuable tool for more cost-effective and efficient in vivo experimental designs in cancer research.

## 5. Patents

The authors have filed a patent application (#10-2024-0181613) describing a method for producing PDCXs using cells obtained from the tumors of ovarian cancer patients, developed in this study.

## Figures and Tables

**Figure 1 cells-14-00325-f001:**
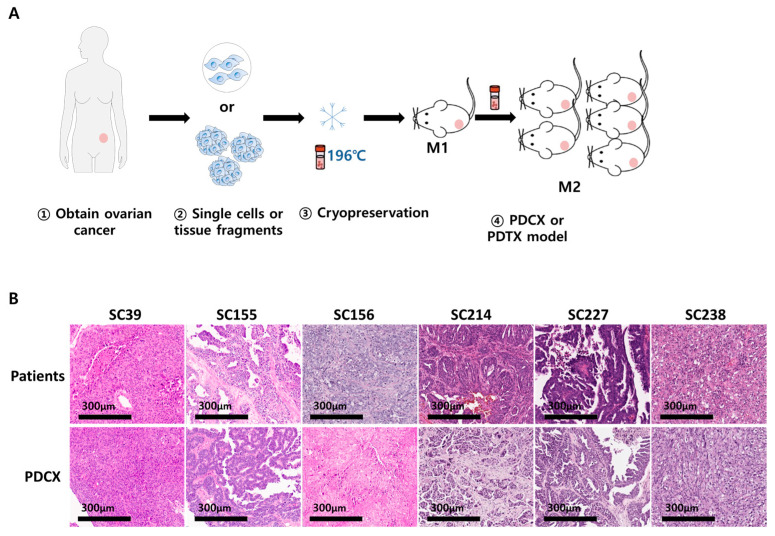
Establishment of PDCX and PDTX models with cryopreserved cells and tissues. (**A**) Modeling process of PDCXs and PDTXs. Fresh primary cancers were dissociated or simply minced prior to cryopreservation. The cryopreserved cancer cells or tissues were transplanted into NSG mice for M1 PDCX or PDTX production. The extracted tumors from M1 PDCXs or PDTXs are processed using the same steps for the next generation fabrication. (**B**) Representative H&E images of M1 PDCX tumors showing the same histological patterns as the primary tumors.

**Figure 2 cells-14-00325-f002:**
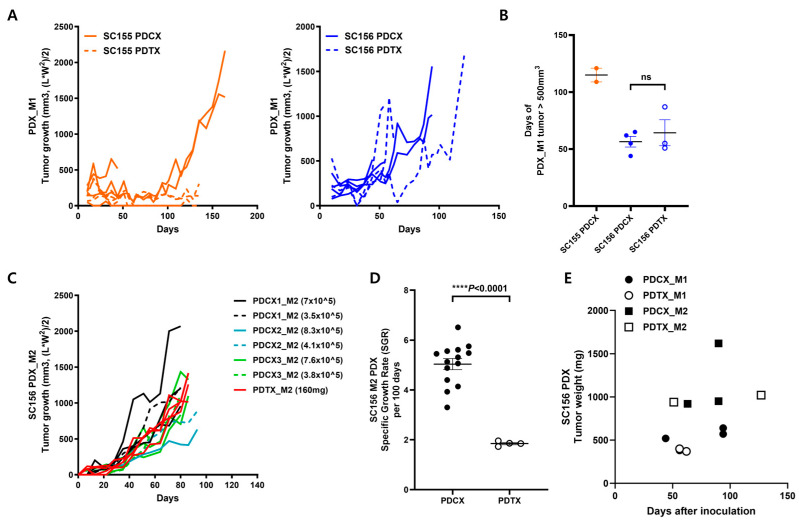
Comparisons of tumor growth between PDCX and PDTX models. (**A**) Tumor growth curves of PDCX_M1 and PDTX_M1 models. Left graph shows SC155 PDCX_M1 (N = 4) and SC155 PDTX_M1 (N = 4), and right graph shows SC156 PDCX_M1 (N = 4) and SC156 PDTX_M1 (N = 3). Each line indicates one tumor growth curve. Solid lines: PDCX; dashed lines: PDTX. (**B**) The number of days until M1 tumor volumes reached 500 mm^3^ for each case (N = 2 to 4). None of the SC155 PDTX tumors grew over 500 mm^3^. The *p*-value is analyzed by unpaired *t*-test function of GraphPad Prism program; *p* = 0.5087. Orange circles indicate SC155 PDCX_M1, Blue circles indicate SC156 PDCX_M1, and Blue empty circles indicate SC156 PDTX_M1. (**C**) Tumor growth curves of SC156 PDCX_M2 and SC156 PDTX_M2. PDCX_M2 tumors with lower cell numbers of implanted cells are shown as dashed lines. PDCX1_M2 (black, solid, N = 3; dashed lines, N = 1), PDCX2_M2 (blue, solid, N = 1; dashed line, N = 1), PDCX3_M2 (green, solid, N = 3; dashed line, N = 1), and PDTX_M2 (solid red line, N = 4) are derived from different M1 entities. The implanted cell numbers are shown in the figures, and each line indicates one tumor growth curve. (**D**) Comparisons of Specific Growth Rates were made between the SC156 M2 PDCX and PDTX tumors from both flanks of M2 PDX mice. The sample size was N = 7 for PDCXs or N = 2 for PDTXs. The *p*-value is analyzed using the unpaired *t*-test function from the GraphPad Prism program; **** *p* < 0.0001. (**E**) The correlation between the tumor weight and extraction days of M1 and M2 tumors, regardless of being PDCXs or PDTXs. This included three M1 PDCXs with their corresponding three M2 PDCXs and two M1 PDTXs with their corresponding two M2 PDTXs.

**Figure 3 cells-14-00325-f003:**
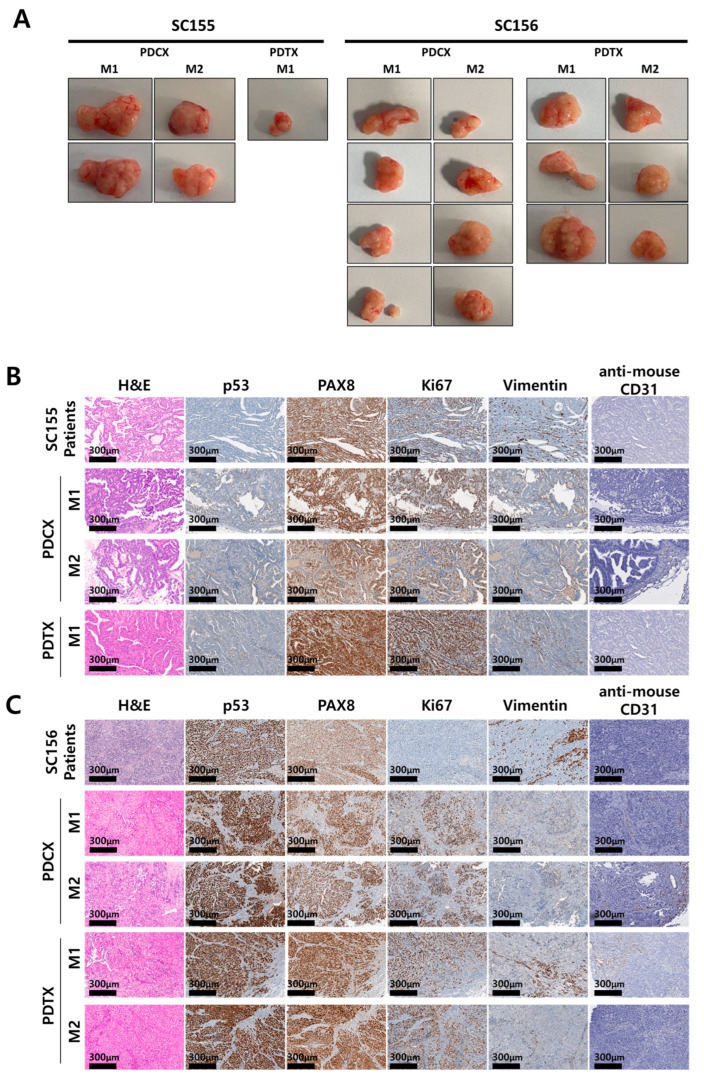
Morphological and histological comparisons between primary cancers and their corresponding PDCX or PDTX tumors in SC155 and SC156. (**A**) Pictures of extracted xenografted tumors from two cases (SC155 and SC156). Two SC155 PDCX_M1 tumors and the corresponding SC155 PDCX_M2 tumors and one SC155 PDTX_M1, which was not passaged to PDTX_M2, are shown. Four SC156 PDCX_M1 tumors and the corresponding SC156 PDCX_M2 tumors, as well as three SC156 PDTX_M1 tumors and the corresponding SC156 PDTX_M2 tumors, are shown. Representative images of H&E and immunohistochemical stains with anti-human p53, PAX8, Ki67, vimentin, and anti-mouse CD31 antibodies for the patients’ primary tumors and their corresponding M1 and M2 tumors for SC155 (**B**) and SC 156 (**C**). The expression of essential markers for ovarian cancer was conserved in M1 and M2 tumors, except for the Ki-67 index in SC156, where M1 and M2 PDCX or PDTX tumors showed a higher proliferation index compared to the patient’s primary tumor.

**Figure 4 cells-14-00325-f004:**
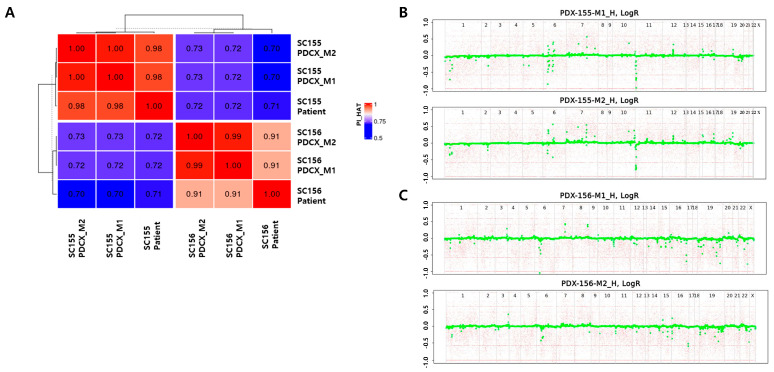
Genetic identities among three generations in two different cases. (**A**) Kinship analysis between the patient’s cancer cells and M1 or M2 PDCX cancer cells. Dark red and high numbers indicate the same person, while dark blue and low numbers indicate different people. (**B**) ASCAT analysis between SC155M1 PDCX and SC155M2 PDCX. (**C**) ASCAT analysis between SC156M1 PDCX and SC156M2 PDCX. ASCAT: allele-specific copy number analysis of tumors. Green dot: copy number alteration (CNA) at a specific gene locus.

**Figure 5 cells-14-00325-f005:**
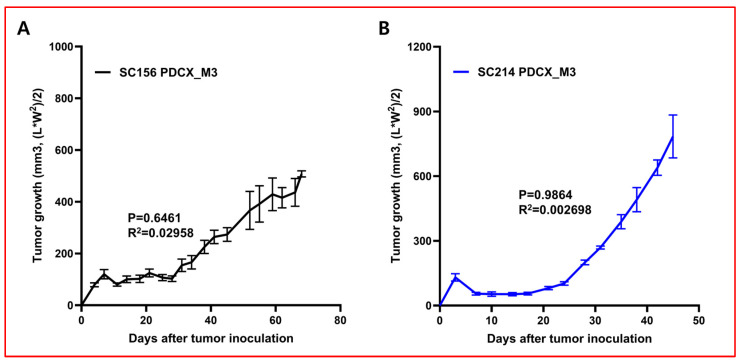
The uniform tumor growth rates of M3 PDCX individuals in two different cases. (**A**) Tumor growth rates of SC156 M3 PDCX mice implanted with 2 × 10^6^ cancer cells (N = 5). One-way ANOVA analysis; *p*-value = 0.6461, R^2^ = 0.02958. (**B**) Tumor growth rates of SC214 M3 PDCX mice implanted with 2 × 10^6^ cancer cells (N = 4). One-way ANOVA analysis; *p*-value = 0.9864, R^2^ = 0.002698. The standard error of the mean (SEM) of the tumor size at each time point is indicated.

**Table 1 cells-14-00325-t001:** Patient information for this study.

Patient ID	Diagnosis	Age	Sex	Clinical Stage	Previous Chemotherapy	Tumor Weight (g)	Number of Tumor Cells
SC18	HGSC	56	F	IIB	No	4.8	7 × 10^7^
SC39	HGSC	56	F	IA	No	2.7	5 × 10^7^
SC101	HGSC	56	F	III	No	2.3	2.4 × 10^8^
SC124	HGSC	43	F	II	No	7.5	1.7 × 10^8^
SC155	HGSC	59	F	IIB	No	11.2	3.2 × 10^8^
SC156	HGSC	68	F	IV	No	6.4	5.2 × 10^7^
SC214	HGSC	33	F	IIIC2	No	18.0	1 × 10^9^
SC227	HGSC	74	F	IV	No	16.4	3.2 × 10^8^
SC236	HGSC	56	F	IIIB	No	15.0	1.58 × 10^8^
SC238	HGSC	62	F	IC	No	13.7	1.5 × 10^8^
SC243	HGSC	46	F	IIA–IIB	No	6.8	2.2 × 10^8^
SC245	HGSC	40	F	IIIC	No	21.4	2.68 × 10^8^
SC246	HGSC	53	F	IC2	No	19.7	4.44 × 10^8^
SC248	HGSC	59	F	IIB	No	24.0	1.57 × 10^8^

**Table 2 cells-14-00325-t002:** Information of primary antibodies in immunohistochemical stain.

Antibody	Company	Cat. No.	Dilution	Retrieval Time	Retrieval Buffer	Antibody Incubation
P53	Dako	71709	1:2000	36 min	EDTA	32 min
PAX8	Roche	760-4618	RTU	36 min	EDTA	32 min
Ki 67	Dako	M7240	1:200	36 min	EDTA	32 min
Vimentin	Zymed	18-0052	1:2000	36 min	EDTA	32 min
Mouse CD31	Cell signaling	77699T	1:1000	30 min	EDTA	60 min

**Table 3 cells-14-00325-t003:** DX establishment with cryopreserved HGSC samples.

Patient ID	Age/Sex	Sample Type	Cryopreserved Days in LN_2_	Amount for Inoculation	Xenograft Success Rate	Xenograft Periods (Days)	Histology Results
SC18	56/F	Cell	557	3 × 10^6^	0/1	N/A	N/A
SC39	56/F	Cell	485	4 × 10^6^	1/1	231	HGSC
SC101	56/F	Cell	269	2 × 10^6^	0/1	N/A	N/A
SC124	43/F	Cell	119	4~10 × 10^6^	0/2	N/A	N/A
SC155	59/F	Tissue	30	150 mg	1/4	193	HGSC
		Cell		5 × 10^6^	2/4	164	HGSC
SC156	68/F	Tissue	19	150 mg	3/3	56~164	HGSC
		Cell		5 × 10^6^	4/4	44~94	HGSC
SC214	33/F	Cell	24/386	4 × 10^6^/5 × 10^6^	3/4	125~310	HGSC, N/A
SC227	74/F	Cell	15	2.5 × 10^6^	2/2	72~183	HGSC
SC236	56/F	Cell	126	7.6 × 10^6^	2/2	270~284	HGSC
SC238	62/F	Cell	79	3.7 × 10^6^	1/1	68	HGSC
SC243	46/F	Cell	37	5.8 × 10^6^	0/2	N/A	N/A
SC245	40/F	Cell	115	1 × 10^6^	1/1	278	HGSC
SC246	53/F	Cell	148	7.5 × 10^6^	0/2	N/A	N/A
SC248	59/F	Cell	100	2.4 × 10^6^	2/2	310	HGSC

**Table 4 cells-14-00325-t004:** Establishment of M2 PDCX and M2 PDTX models.

Sample Type	Mouse ID	Days of Tumor Growth	Left Flank			Right Flank			Average SGR per 100 Days
			Injected Amount	Extracted Amount	SGR per 100 Days	Injected Amount	Extracted Amount	SGR per 100 Days	
Tissue	SC156T4M2-1	87	160 mg	800 mg	1.85	160 mg	720 mg	1.73	1.85
	SC156T4M2-2	87	160 mg	880 mg	1.96	160 mg	810 mg	1.86	
Cell	SC156C1M2-1	80	7 × 10^5^	670 mg (4.88 × 10^7^)	5.31	7 × 10^5^	1140 mg (6.97 × 10^7^)	5.75	5.04
	SC156C1M2-2	80	3.5 × 10^5^	1190 mg (6.4 × 10^7^)	6.51	7 × 10^5^	970 mg (6.23 × 10^7^)	5.61	
	SC156C2M2-1	104	4 × 10^5^	660 mg (2.96 × 10^7^)	4.14	8.3 × 10^5^	530 mg (2.57 × 10^7^)	3.3	
	SC156C3M2-1	87	7.6 × 10^5^	700 mg (5.29 × 10^7^)	4.88	7.6 × 10^5^	1200 mg (8.74 × 10^7^)	5.45	
	SC156C3M2-2	87	3.8 × 10^5^	540 mg (3.11 × 10^7^)	5.06	7.6 × 10^5^	580 mg (3.48 × 10^7^)	4.40	
	SC156C3M2-3	80	4 × 10^5^	400 mg (0.93 × 10^7^)	3.93	4 × 10^5^	1050 mg (3.2 × 10^7^)	5.48	
	SC156C3M2-4	86	4 × 10^5^	680 mg (3.31 × 10^7^)	5.13	4 × 10^5^	1000 mg (4.79 × 10^7^)	5.56	

**Table 5 cells-14-00325-t005:** Comparison of immunohistochemical markers between original human tumors and M1 and M2 generation of PDCX and PDTX models.

		P53	PAX8	Ki67	Vimentin	Anti-Mouse CD31
SC155	Primary tumor	Null, (−)	100% (+)	70~90%	(+) in stroma and b.v.	(−)
	PDCX M1	Null, (−)	100% (+)	80~90%	(+) in stroma, focally, and b.v.	(+) in b.v.
	PDCX M2	Null, (−)	100% (+)	80~90%	(+) in b.v.	(+) in b.v.
	PDTX M1	Null, (−)	100% (+)	80~90%	(+) in b.v.	(+) in b.v.
SC156	Primary tumor	100% (+)	100% (+)	15~25%	(+) in stroma, focally, and b.v.	(−)
	PDCX M1	100% (+)	100% (+)	80~90%	(+) in stroma, focally, and b.v.	(+) in b.v.
	PDCX M2	100% (+)	100% (+)	80~90%	(+) in b.v.	(+) in b.v.
	PDTX M1	100% (+)	100% (+)	80~90%	(+) in stroma, focally, and b.v.	(+) in b.v.
	PDTX M2	100% (+)	100% (+)	80~90%	(+) in b.v.	(+) in b.v.

b.v.: blood vessels.

**Table 6 cells-14-00325-t006:** Genetic identity between primary tumor and corresponding M1 and M2 tumors.

IID1	IID2	RT	EZ	Z0	Z1	Z2	PI_HAT	PHE	DST	PPC	RATIO
SC155	PDX-155-M1	UN	NA	0.0000	0.0439	0.9561	0.9780	−1	0.988684	1.0000	NA
SC155	PDX-155-M2	UN	NA	0.0000	0.0445	0.9555	0.9777	−1	0.988527	1.0000	NA
PDX-155-M1	PDX-155-M2	UN	NA	0.0000	0.0090	0.9910	0.9955	−1	0.997669	1.0000	NA
SC156	PDX-156-M1	UN	NA	0.0000	0.1837	0.8163	0.9081	−1	0.952665	1.0000	NA
SC156	PDX-156-M2	UN	NA	0.0000	0.1825	0.8175	0.9088	−1	0.952985	1.0000	NA
PDX-156-M1	PDX-156-M2	UN	NA	0.0000	0.0109	0.9891	0.9946	−1	0.997202	1.0000	NA

## Data Availability

There are no publicly accessible data available.

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
