# Peer review of "Establishment and Its Utility of a Patient-Derived Cell Xenografts (PDCX) Model with Cryopreserved Cancer Cells from Human Tumor"

_cells, 2025, doi:10.3390/cells14050325_

Round 1
Reviewer 1 Report
Comments and Suggestions for Authors
The manuscript of Kim et al. reports on a novel method to generate a patient-derived xenograft (PDX) model of ovarian cancer (high grade serous ovarian cancer) in immunodeficient mice by using cryopreserved cancer cells instead of patient derived tissue samples. They show that primary cells have a high engraftment rates of about 64% instead of 58 % and the technical feasibility is easier compared to tissue engraftment method. The results are promising and the explanation of the method is clear. However, the structure of described results and the overall novelty of this method is not clear. There are several prior papers which have utilized ovarian cancer for PDX engraftments and also with frozen tissues. Data quality: the quality of the histologic images is quite low and technical concerns come to mind. Thus, I would not recommend the paper for publication in cells journal.
Major
· Patient Enrollment and Sample Details: It is unclear how many patients were enrolled in the study and how many tumor samples were collected. Furthermore, the patient characteristics (e.g., age, stage of cancer, treatment history) are entirely missing. This lack of information makes it difficult to assess the representativeness and relevance of the study cohort.
· Figure 1b: The image quality is too low, and the images are too small. Higher resolution images and improved formatting are needed to make the results interpretable.
· Figure 2a and 2c: These figures should be optimized for clarity. Currently, there are too many lines in the diagram, making it difficult for readers to interpret the data.
· The authors indicate that tumor cells can be used for PDCX engraftment within 15 to 557 days of cryopreservation. However, the rationale for selecting these time frames is not provided, and the results related to these findings are not presented in the manuscript. Clarification and supporting data are required.
· The authors report a high tumor implantation success rate of about 64% using their cell implantation method. However, additional information on how this efficiency compares to other methods (e.g., fresh tissue implantation) is needed to contextualize the significance of these findings. A Bland-Altman abalysis would be helpful to compare the both methods
· The authors mention that uniform-sized tumors were formed in the third-generation PDCX model using a small number of cells. However, the specific criteria for defining "uniform-sized tumors" and the variability between models are not discussed in detail.
· The argument that laboratories may face challenges in performing surgical techniques (e.g., anesthesia, incision, suturing) for traditional PDX models is weak. Any laboratory using animals for research purposes should already be equipped with the necessary materials and expertise to perform these procedures safely and ethically. Simplifying this aspect does not sufficiently justify the proposed method’s superiority.
Minor Comments
The manuscript requires significant improvement in English language usage. For example:
Line 87: “However, in most of reports covering PDCX, the mice were injected using cancer cells that had never been frozen. Therefore, the experiments must have been laborious, as many steps had to be completed immediately within a short period of time.”
Author Response
Reviewer 1
The manuscript of Kim et al. reports on a novel method to generate a patient-derived xenograft (PDX) model of ovarian cancer (high grade serous ovarian cancer) in immunodeficient mice by using cryopreserved cancer cells instead of patient derived tissue samples. They show that primary cells have a high engraftment rates of about 64% instead of 58 % and the technical feasibility is easier compared to tissue engraftment method. The results are promising and the explanation of the method is clear. However, the structure of described results and the overall novelty of this method is not clear. There are several prior papers which have utilized ovarian cancer for PDX engraftments and also with frozen tissues. Data quality: the quality of the histologic images is quite low and technical concerns come to mind. Thus, I would not recommend the paper for publication in cells journal.
Major
Comments 1. Patient Enrollment and Sample Details: It is unclear how many patients were enrolled in the study and how many tumor samples were collected. Furthermore, the patient characteristics (e.g., age, stage of cancer, treatment history) are entirely missing. This lack of information makes it difficult to assess the representativeness and relevance of the study cohort.
Response 1. In response to the reviewer’s comments, we added the number of patients (line 102) and Table 1 (line 111) in the Materials and Methods section. This table includes clinico-pathological information on the 14 patients who participated in the study, including age, cancer stage, and history of previous chemotherapy.
Comments 2. Figure 1b: The image quality is too low, and the images are too small. Higher resolution images and improved formatting are needed to make the results interpretable.
Response 2. As the reviewer recommended, we increased the size of all H&E staining images in Figure 1B and any low-resolution images were replaced with high-quality scanned versions.
Comments 3. Figure 2a and 2c: These figures should be optimized for clarity. Currently, there are too many lines in the diagram, making it difficult for readers to interpret the data.
Response 3. In response to the reviewer’s comments, we have made a new version of these graphs to enhance the readability. For Figure 2a, which compares tumor growth in PDCX and PDTX models, we have separated the data for SC155 and SC156 specimens into two distinct graphs (left: SC155, right: SC156), using solid lines for PDCX and dashed lines for PDTX. In Figure 2c, comparing tumor growth curves of PDCX_M2 with various cell numbers and PDTX_M2 with 160 mg tumor implants, we used dashed lines for lower(half) cell number implants and different colors to distinguish between groups (3 PDCX groups - black, blue, green lines; 1 PDTX group - red). We think these modifications would make it easier for readers to discern the different growth patterns. We added detailed explanations in the figure legends.
Comments 4. The authors indicate that tumor cells can be used for PDCX engraftment within 15 to 557 days of cryopreservation. However, the rationale for selecting these time frames is not provided, and the results related to these findings are not presented in the manuscript. Clarification and supporting data are required.
Response 4. As the reviewer mentioned, this wide range of cryopreservation periods allowed us to observe PDCX engraftment efficiency across various preservation durations, as shown in Table S2.
When the cryopreservation periods are divided into three groups, short (under 100 days), middle (between 100 days and 1 year), and long (over 1 year), the engraftment rates were 73.3% in short-term, 50% in middle-term, and 50% in long-term. These results demonstrate the stability of cryopreserved cancer cells for PDCX establishment after long-term storage, despite a slight decrease in engraftment rates for middle and long cryopreservation periods.
These rates are substantially higher than previously reported research, which showed a 9% engraftment rate when establishing PDX models with tissues cryopreserved for over 52 weeks (Ivanics, T. et al., 2018). This observation suggests that our cryopreservation method can maintain cell viability for extended periods, which is valuable for long-term biobanking and flexibility in experimental planning. To address this feedback further, we have added discussions related to these findings in the Result section (line 307) and Discussion section (line 582).
Table S2. Engraftment rates of PDCX_M1 according to the cryopreservation periods
|
Duration |
Case |
Cryopreserved Period |
Engraftment |
Engraftment rate |
|
Short (< 100 days) |
SC155 |
30 |
2/4 |
11/15 (73.3%) |
|
SC156 |
19 |
4/4 |
||
|
SC214 |
24 |
2/2 |
||
|
SC227 |
15 |
2/2 |
||
|
SC238 |
79 |
1/1 |
||
|
SC243 |
37 |
0/2 |
||
|
Middle (100 ~ 365 days) |
SC101 |
269 |
0/1 |
5/10 (50%) |
|
SC124 |
119 |
0/2 |
||
|
SC236 |
126 |
2/2 |
||
|
SC245 |
115 |
1/1 |
||
|
SC246 |
148 |
0/2 |
||
|
SC248 |
100 |
2/2 |
||
|
Long (365 days <) |
SC18 |
557 |
0/1 |
2/4 (50%) |
|
SC39 |
485 |
1/1 |
||
|
SC214 |
386 |
1/2 |
Comments 5. The authors report a high tumor implantation success rate of about 64% using their cell implantation method. However, additional information on how this efficiency compares to other methods (e.g., fresh tissue implantation) is needed to contextualize the significance of these findings. A Bland-Altman analysis would be helpful to compare the both methods
Response 5. We appreciate the reviewer’s comments for a more comprehensive comparison between our cell implantation method (PDCX) and the fresh tissue implantation method (PDTX).
To further contextualize our findings, we conducted a Bland-Altman analysis using GraphPad Prism program, as the reviewer recommended. This analysis clearly demonstrated a substantial difference in tumor growth rates between PDCX and PDTX models, which we have included as Supplemental Figure 1.
Ideally, we would have compared tumor weights of PDCX to PDTX after implanting equal amounts in both models. However, since we did not use equivalent quantities for implantation in this study, we compared the Specific Growth Rate (SGR) values between these two groups. As shown in Figure 2D, we compared the SGR values of SC156 PDCX_M2 and SC156 PDTX_M2 at the 100-day mark. The analysis revealed that PDCX_M2 exhibited significantly faster tumor growth compared to PDTX_M2 (5.04 vs. 1.85).
In Bland-Altman analysis, if the plot shows a pattern where the difference increases or decreases depending on the measurement value's size, it supports inconsistency between the two methods. Our results showed that the 95% limits of agreement did not include 0 (ranging from 1.591 to 5.129), and the graph displayed a positive correlation between the average (mean of SGR) and the difference (standard deviation). These findings further substantiate the significant differences between PDCX and PDTX models.
We have added these details to line 406 of the manuscript, enhancing the comprehensive nature of our analysis and providing a more robust comparison between the two xenograft models.
Comments 6. The authors mention that uniform-sized tumors were formed in the third-generation PDCX model using a small number of cells. However, the specific criteria for defining "uniform-sized tumors" and the variability between models are not discussed in detail.
Response 6. To address this concern and provide a more rigorous analysis of tumor size consistency, we modified the graphs in Figure 5 to display grouped growth curve instead of individual growth curves and implemented the Ordinary One-way ANOVA method. This statistical approach considers differences between groups as significant when the P-value is below 0.05 and the R-square value approaches 1. Conversely, P-values closer to 1 and R-square values closer to 0 indicate minimal differences between subjects. Upon analyzing the tumor growth curves of PDCX_M3 shown in Figure 5, we found that for SC156 PDCX_M3 (5 animals), the P-value was 0.6461 and the R-square was 0.02958, indicating no significant difference in tumor growth curves. For SC214 PDCX_M3 (4 animals), we observed a P-value of 0.9864 and an R-square of 0.002698, suggesting nearly identical growth patterns. These results provide quantitative support for our claim of uniform-sized tumors in the third-generation PDCX model, particularly demonstrating high consistency in tumor growth across individual animals within each group. We have added these details to the subsection 3.5 in Results section (line 533).
Comments 7. The argument that laboratories may face challenges in performing surgical techniques (e.g., anesthesia, incision, suturing) for traditional PDX models is weak. Any laboratory using animals for research purposes should already be equipped with the necessary materials and expertise to perform these procedures safely and ethically. Simplifying this aspect does not sufficiently justify the proposed method’s superiority.
Response 7. I agree with the reviewer’s opinion. However, the injecting cells is easier than performing surgical procedure, even for laboratories experienced in surgical techniques. Moreover, PDCX modeling might reduce the surgical burden on mice. This approach not only simplifies the process but also potentially improves animal welfare by minimizing invasive procedures. We have addressed these points in the manuscript, in the discussion section at line 654, where we elaborate on the comparative advantages of the PDCX method in terms of technical simplicity and reduced animal stress.
Minor Comments
Comments 1. The manuscript requires significant improvement in English language usage. For example:
Line 87: “However, in most of reports covering PDCX, the mice were injected using cancer cells that had never been frozen. Therefore, the experiments must have been laborious, as many steps had to be completed immediately within a short period of time.”
Response 1. This paper will be proofread in English after revision.
Reviewer 2 Report
Comments and Suggestions for Authors
This manuscript presents a novel patient-derived cell xenograft (PDCX) model using cryopreserved cancer cells from high-grade serous ovarian cancer (HGSC). The authors claim that the PDCX model offers advantages over traditional PDX models, such as higher engraftment efficiency, stable cryopreservation, and uniform tumor sizes among animals. The study demonstrates that the PDCX model maintains the morphological and molecular characteristics of the original tumors and shows a faster growth rate compared to the PDTX model. The authors suggest that the PDCX model could be a valuable tool for personalized cancer treatment evaluation. Overall, the manuscript presents an innovative approach to cancer modeling. However, the following issues are required for explaining:
1. It is essential to validate the preservation of immune cells and stromal cells within the PDCX model derived from human HGSC. Techniques such as single-cell sequencing and immunofluorescence should be employed to verify the integrity and functionality of these cells.
2. The authors should investigate whether the PDCX model can predict chemotherapy drug sensitivity in both in vivo and in vitro experiments.
3. The authors should include photographs of tumor formation in mice, comparing PDCX and PDTX models. Additionally, images of excised tumors should be provided to visually substantiate the engraftment efficiency and tumor morphology.
4. For cryopreserved cancer samples, the authors should investigate if immunotherapy-related biomarkers (e.g., PD-L1, TMB) remain consistent between PDCX and PDTX models. This consistency is crucial for determining the efficacy of immunotherapy treatments.
5. A more comprehensive description of the experimental protocol is needed. Detailed steps and conditions should be included to ensure reproducibility and to clarify the experimental outcomes.
6. The potential of the PDCX model to serve as a predictive tool for drug sensitivity should be thoroughly discussed. The authors should highlight how the model could be integrated into personalized treatment plans and its advantages over existing models.
7. Some of the supplementary figures should be moved to the main manuscript to provide a clearer and more comprehensive presentation of the data.
8. The authors should add a Figure (or Graphical abstract) to summarize the finding of this study.
9. Essential details of data and analysis in this study remain unclear. For example, there is a lack of details on how Copy Number Alteration data were processed and normalized? A reproducible detail should be provided.
10. Figure 1E: Scale bars should be provided.
11. Multi-comparison of One-Way ANOVA: does the adjusted p value showed in figures?
12. The authors are recommended to consider engaging a professional language editing service to ensure the clarity and coherence of the manuscript.
Comments on the Quality of English Language
The English could be improved to more clearly express the research.
Author Response
Reviewer 2
This manuscript presents a novel patient-derived cell xenograft (PDCX) model using cryopreserved cancer cells from high-grade serous ovarian cancer (HGSC). The authors claim that the PDCX model offers advantages over traditional PDX models, such as higher engraftment efficiency, stable cryopreservation, and uniform tumor sizes among animals. The study demonstrates that the PDCX model maintains the morphological and molecular characteristics of the original tumors and shows a faster growth rate compared to the PDTX model. The authors suggest that the PDCX model could be a valuable tool for personalized cancer treatment evaluation. Overall, the manuscript presents an innovative approach to cancer modeling. However, the following issues are required for explaining:
Comments 1. It is essential to validate the preservation of immune cells and stromal cells within the PDCX model derived from human HGSC. Techniques such as single-cell sequencing and immunofluorescence should be employed to verify the integrity and functionality of these cells.
Response 1. We appreciate the suggestion to validate immune and stromal cell preservation in our PDCX model.
In this study, we used immunohistochemistry (IHC) to verify the integrity of stromal cells with vimentin antibody and examine p53, PAX8, Ki67, expression, which are essential molecular markers for HGSC diagnosis. Additionally, we conducted whole-exome sequencing (WES) analysis, which demonstrated that tumors from M1 and M2 PDCX preserve their genetic characteristics from the corresponding patients’ primary tumors and maintain similar copy number variation (CNV) patterns across three generations.
However, human immune cells could not be preserved in PDCX models even in immunodeficient mice. Consequently, humanized PDTX or PDCX models should be developed to mimic the patient’s tumor immune microenvironments more accurately. Therefore, we did not perform the studies for immune cell status in our PDCX models.
We believe our comprehensive analyses sufficiently validate the PDCX model's cellular and genetic integrity, balancing research thoroughness with resource efficiency.
Comments 2. The authors should investigate whether the PDCX model can predict chemotherapy drug sensitivity in both in vivo and in vitro experiments.
Response 2. We appreciate the reviewer's suggestion to investigate the PDCX model's ability to predict chemotherapy drug sensitivity in both in vivo and in vitro experiments. Our PDCX models could be widely employed for various research applications previously conducted with PDTX models, including drug screening, preclinical studies for therapeutic effects, and drug resistance mechanisms. Based on the available literature, we can address this concern as follows: 1) In vivo prediction potential: PDCX models, like PDX models, can reproduce drug responses observed in patients. For example, a study showed concordant effects of 5-fluorouracil and radiotherapy in colorectal cancer patients and PDX models (Liu, Y.H. et al., 2023). 2) In vitro prediction potential: While our current study focuses on in vivo applications, PDCX models could be adapted for in vitro drug sensitivity predictions, similar to recent successes with patient-derived organoids and cell cultures for drug screening (Tang, Y.C. et al., 2022). We have added about these details to the Discussion section (line 654).
Comments 3. The authors should include photographs of tumor formation in mice, comparing PDCX and PDTX models. Additionally, images of excised tumors should be provided to visually substantiate the engraftment efficiency and tumor morphology.
Response 3. Unfortunately, we did not take a picture of tumors comparing PDCX and PCTX models. However, we took pictures for the extracted tumors from PDCX and PDTX models, which were added to Figure 3.
Comments 4. For cryopreserved cancer samples, the authors should investigate if immunotherapy-related biomarkers (e.g., PD-L1, TMB) remain consistent between PDCX and PDTX models. This consistency is crucial for determining the efficacy of immunotherapy treatments.
Response 4. We appreciate the reviewer's interest in biomarkers related to immunotherapy and specific gene analyses such as TMB or PD-L1. However, our study primarily focused on developing the PDCX model and did not delve deeply into these aspects. Unfortunately, we did not anticipate specific gene or tumor mutation burden (TMB) analysis during our initial WES run. For specific analysis like TMB, customized setting is necessary at the initial processing stage. Currently, we face constraints in terms of available samples and time, which prevent us from conducting a new analysis specifically for TMB.
We hope the reviewer understand our situation. Nevertheless, the WES analysis we performed provides a comprehensive view of the overall gene patterns. As shown in Figure 4, our results indicate no significant differences in the overall genetic profiles between patients, PDCX, and PDTX models.
Comments 5. A more comprehensive description of the experimental protocol is needed. Detailed steps and conditions should be included to ensure reproducibility and to clarify the experimental outcomes.
Response 5. We appreciate the reviewer's feedback regarding the need for a more comprehensive description of the experimental protocol. In response, we have revised the methods section (line 132) to include detailed steps for each protocol, organized with numbered lists to enhance clarity and ensure reproducibility. This structured approach allows readers to easily follow the procedures and understand the experimental outcomes.
Comments 6. The potential of the PDCX model to serve as a predictive tool for drug sensitivity should be thoroughly discussed. The authors should highlight how the model could be integrated into personalized treatment plans and its advantages over existing models.
Response 6. A new paragraph was added to the Discussion section (line 654-678) regarding the potential for future experiments that could be performed using the PDCX model.
Comments 7. Some of the supplementary figures should be moved to the main manuscript to provide a clearer and more comprehensive presentation of the data.
Response 7. We moved the supplemental figure to the main manuscript as Figure 3A.
Comments 8. The authors should add a Figure (or Graphical abstract) to summarize the finding of this study.
Response 8. A new Graphical abstract was added to the revised manuscript.
Comments 9. Essential details of data and analysis in this study remain unclear. For example, there is a lack of details on how Copy Number Alteration data were processed and normalized? A reproducible detail should be provided.
Response 9. We have provided a detailed explanation of the overall genetic analysis, including how the Copy Number Alteration (CNA) data were processed and normalized in the "Genetic Analysis" section of the Materials and Methods.
Comments 10. Figure 1E: Scale bars should be provided.
Response 10. We added scale bars to Figure 1B and Figure 3.
Comments 11. Multi-comparison of One-Way ANOVA: does the adjusted p value showed in figures?
Response 11. To clarify, we did not perform multi-comparison tests to obtain adjusted p-values. All results presented in the manuscript were derived from single comparisons using the One-way ANOVA function in GraphPad Prism software, which analyzes the general p-values.
Comments 12. The authors are recommended to consider engaging a professional language editing service to ensure the clarity and coherence of the manuscript.
Response 12. This paper will be proofread in English after revision.
Reviewer 3 Report
Comments and Suggestions for Authors
This manuscript entitled "Establishment and its Utility of a Patient-Derived Cell Xenografts (PDCX) Model with Cryopreserved Cancer Cells from Human Tumor" by Kim et al. investigated the impact of cryopreservation duration on the efficiency of generating PDCX model. From the data the author presented, they claimed the cryopreservation time can be up to 485 days. However, the sample size of xenografts tested in this manuscript is low, especially for the longer duration of cryopreservation. In my opinion, it’s not statistically significant to drive a definitive conclusion that duration of cryopreservation affected the efficiency of generating PDTX or PDCX model. the data presented may be not that convincing to drive a definitive conclusion. I would advise the authors expand the numbers tested for different duration time and examine the correlation between cryopreservation time and efficiency of generation PDCX model.
Some specific comments that would improve the quality of the manuscript.
Major comments:
I would advise the authors expand the numbers tested for different duration time and examine the correlation between cryopreservation time and efficiency of generation PDCX model.
Minor comments:
1. Abstract: It should be precise description of conclusions. Also, I would advise the authors not to show “****p<0.00001” in the abstract.
2. Fig. 1b should provide scale bar for each panel.
3. It’s not clear why the authors showed two different cryopreserved days for SC214.
4. Fig. 2, lack of detailed description of data, and lack of error bar information. Which kind of test to generate the p value? The information should be provided the legends.
5. Fig. 3 should provide scale bar for each panel.
6. Fig. 5b, it’s not clear each data point represents mean or median. And the data lack of SD or SEM information.
Author Response
Reviewer 3
This manuscript entitled "Establishment and its Utility of a Patient-Derived Cell Xenografts (PDCX) Model with Cryopreserved Cancer Cells from Human Tumor" by Kim et al. investigated the impact of cryopreservation duration on the efficiency of generating PDCX model. From the data the author presented, they claimed the cryopreservation time can be up to 485 days. However, the sample size of xenografts tested in this manuscript is low, especially for the longer duration of cryopreservation. In my opinion, it’s not statistically significant to drive a definitive conclusion that duration of cryopreservation affected the efficiency of generating PDTX or PDCX model. the data presented may be not that convincing to drive a definitive conclusion. I would advise the authors expand the numbers tested for different duration time and examine the correlation between cryopreservation time and efficiency of generation PDCX model.
Some specific comments that would improve the quality of the manuscript.
Major comments:
Comments 1. I would advise the authors expand the numbers tested for different duration time and examine the correlation between cryopreservation time and efficiency of generation PDCX model.
Response 1. In this study, a total of 14 cases were utilized for PDCX generation. As shown in the following table (Table S2), the cases were categorized according to the duration of cryopreservation time and assessed for engraftment efficiency.
Table S2. Engraftment rates of PDCX_M1 according to the cryopreservation periods
|
Duration |
Case |
Cryopreserved Period |
Engraftment |
Engraftment rate |
|
Short (< 100 days) |
SC155 |
30 |
2/4 |
11/15 (73.3%) |
|
SC156 |
19 |
4/4 |
||
|
SC214 |
24 |
2/2 |
||
|
SC227 |
15 |
2/2 |
||
|
SC238 |
79 |
1/1 |
||
|
SC243 |
37 |
0/2 |
||
|
Middle (100 ~ 365 days) |
SC101 |
269 |
0/1 |
5/10 (50%) |
|
SC124 |
119 |
0/2 |
||
|
SC236 |
126 |
2/2 |
||
|
SC245 |
115 |
1/1 |
||
|
SC246 |
148 |
0/2 |
||
|
SC248 |
100 |
2/2 |
||
|
Long (365 days <) |
SC18 |
557 |
0/1 |
2/4 (50%) |
|
SC39 |
485 |
1/1 |
||
|
SC214 |
386 |
1/2 |
When the cryopreservation periods were divided into three period groups, short (under 100 days), middle (between 100 days and 1 year), and long (over 1 year), the engraftment rates were 73.3% for short-term, 50% for middle-term, and 50% for long-term. This result demonstrates the stability of cryopreserved cancer cells for PDCX establishment after long-term storage exceeding 1 year, although the rate slightly decreased in the cases with middle and long period. This observation has significant implications for long-term biobanking and offers flexibility in experimental design and execution.
To address this feedback further, we have added discussions related to these findings in the Result section (line 307) and the Discussion section (line 582).
Minor comments:
Comments 1. Abstract: It should be precise description of conclusions. Also, I would advise the authors not to show “****p<0.00001” in the abstract.
Response 1. The p-value indicated in the abstract has been deleted.
Comments 2. Fig. 1b should provide scale bar for each panel.
Response 2. We have added scale bars to Figure 1B.
Comments 3. It’s not clear why the authors showed two different cryopreserved days for SC214.
Response 3. We appreciate the reviewer's question regarding the two different cryopreservation durations for case SC214. In this instance, we were able to obtain a sufficient amount of tissue sample. Consequently, we first generated the PDCX model after a short cryopreservation period of 24 days. To further validate the effectiveness of PDCX generation after an extended duration, we conducted a subsequent experiment after 386 days of cryopreservation. This rationale has been clearly stated in the Results section (subsection 3.1 on line 307) of the manuscript.
Comments 4. Fig. 2, lack of detailed description of data, and lack of error bar information. Which kind of test to generate the p value? The information should be provided the legends.
Response 4. We appreciate the reviewer's feedback regarding Figure 2. In response, we have clarified that for panels 2B and 2D, the p-values were generated using the Unpaired t-test function in GraphPad Prism software. Additionally, we have included detailed descriptions of the data and added information about the error bars in the figure legend.
Comments 5. Fig. 3 should provide scale bar for each panel.
Response 5. We have added scale bars to Figure 3.
Comments 6. Fig. 5b, it’s not clear each data point represents mean or median. And the data lack of SD or SEM information.
Response 6. We appreciate the reviewer's comments regarding Figure 5. To clarify, Figure 5 illustrates the tumor growth curves for SC156 PDCX_M3 (Figure 5A) and SC214 PDCX_M3 (Figure 5B), which represent data from five and four mice, respectively. To enhance clarity and address potential confusion, we have grouped the tumor growth curves for each mouse case into a single line. Additionally, we have included the SEM (standard error of the mean) values at each time point where tumor size was measured. This information has also been added to the figure legend.
Round 2
Reviewer 1 Report
Comments and Suggestions for Authors
Thank you for the opportunity to review your manuscript. I sincerely recognize and appreciate the effort you have invested in addressing the previous feedback and revising your work. While I commend your dedication and the improvements made, I remain of the opinion that the manuscript does not align with the scope or quality standards expected of Cells Journal.
Reviewer 2 Report
Comments and Suggestions for Authors
The revised manuscript has made a great improvement. I have no more comments and recommends.
Reviewer 3 Report
Comments and Suggestions for Authors
I have no further comments.